# Translation Memory Guided Neural Machine Translation

## Abstract

Many studies have proven that Translation Memory (TM) can help improve the translation quality of neural machine translation (NMT). Existing ways either employ extra encoder to encode information from TM or concatenate source sentence and TM sentences as encoder's input. These previous methods don't model the semantic relationship between the source sentence and TM sentences. Meanwhile, the training corpus related to TM is limited, and the sentence level retrieval approach further limits its scale. In this paper, we propose a novel method to combine the strengths of both TM and NMT. We treat the matched sentence pair of TM as the additional signal and apply one encoder enhanced by the pre-trained language model (PLM) to encode the TM information and source sentence together. Additionally, we extend the sentence level retrieval method to the n-gram retrieval method that we don't need to calculate the similarity score. Further, we explore new methods to manipulate the information flow from TM to the NMT decoder. We validate our proposed methods on a mixed test set of multiple domains. Experiment results demonstrate that the proposed methods can significantly improve the translation quality and show strong adaptation for an unknown or new domain.

## 1 Introduction

Neural machine translation (NMT), an end-to-end approach, has achieved state-of-the-art translation performance on many language pairs (Vaswani et al., 2017; Wang et al., 2019). Usually, a trained NMT model translates a new sentence into the target language from scratch. However, human translators can quickly and accurately translate a sentence by reuse existing repetitive translation fragments in the translation memory (TM). Therefore, we naturally think of using TM to improve the translation quality of NMT. Typically, a TM consists of bilingual parallel sentence pairs(TM-source and TM-target) that are similar to the current sentence to be translated (Koehn & Senellart, 2010; Cao & Xiong, 2018). And from statistical machine translation (SMT) to NMT, a variety of efforts have been made to integrate a TM into machine translation.

The process of integrating TM information and NMT mainly includes two steps: TM retrieval and fusion of TM information and NMT network. For the fusion of TM and NMT, such attempts have already been conducted. And a commonly used integration way is to employ the multi-encode structure. Cao & Xiong (2018) propose a simple method that employs a new encoder to encode the TM information to guide the decoding process and Xia et al. (2019) also use a graph-based encoder to pack the TM sentences into a graph. Their methods all require additional encoder structure, ignore the TM-source information, and only encode the TM-target information. These will cause a series of problems. On the one hand, it will significantly increase the parameter scale of the network. On the other hand, the encoding process of TM-target information and source sentence are isolated from each other, so the semantic connection between them is lost.

About TM retrieval, various metrics can be used to estimate the similarity score of two sentences. We select the sentences with the highest similarity from the TM database for the current source sentence by calculating the similarity score. The existing retrieval approaches used in previous work are usually to calculate the sentence level similarity score, such as Edit-Distance (Gu et al., 2017; Xia et al., 2019), IDF-based similarity score (Bapna & Firat, 2019), and cosine similarity (Xu et al., 2020). TM's current work is experimenting with relatively small data sets that are usually only hundreds of thousands of sentences. One main reason is that we use sentence-level similarity. When

we also set a relatively high similarity threshold for a source sentence, we have a high probability that we will not find a similar sentence in the TM database. Although Bapna & Firat (2019) and Xu et al. (2020) also use the n-gram method to search, they still need to select the corresponding sentence that maximizes the n-gram similarity with the source sentence. Meanwhile, a small training set will also lead to insufficient training of network parameters.

In this paper, to address the problem presented above, we proposed a novel and effective method for the combination of TM and NMT. The proposed approach's key idea is to treat the matched TM-source and TM-target as the additional signal and try to encode them with the source sentence together. Specifically, we first find the matched TM-source and TM-target sentence pairs from our training corpus. To enhance the semantic relationship between source sentence, TM-source, and TM-target, we use a universal encoder to encode the three sentences and obtain context representation information simultaneously. Then we explore and try four methods to incorporate the context information into the decoding network. To further strengthen the ability to copy useful information from TM context information and alleviate the rare-word problem, we integrate pointer network (Gulcehre et al., 2016; Gu et al., 2016; See et al., 2017) into the decoder.

To obtain sufficient training corpus and train network parameters more fully and effectively, in this paper, we also modify the retrieval algorithm and use a pre-trained language model (PLM) to initialize the encoder's parameters. Partially inspired by phrase-based SMT, we don't compute the sentence level similarity score between two sentences in our retrieved method. If two sentences have a common n-gram segment, we assume that they are similar, and the sentence pairs of the TM database can provide a useful segment to help improve the translation quality. Currently, many studies have proven that PLM can offer valuable prior knowledge to enhance the translation performance of NMT (Weng et al., 2020; Song et al., 2019), So we also employ PLM to initialize the parameters of the encoder and give encoder well-trained parameters as a starting point.

To validate the proposed approach's effectiveness, we implement our idea on top of the state-of-the-art model Transformer (Vaswani et al., 2017). A series of experiments on the English-to-French translation task demonstrate that the proposed method can significantly improve NMT with the TM information. In summary, we make three main contributions:

- We employ the n-gram retrieval to find a similar sentence, and this is very simple and fast. It does not need the complicated fuzzy matches algorithm to calculate the similarity between the source sentence and TM-source from TM or training data.

- Does not need an extra encoder to encode the retrieved sentence from TM and use only an encoder enhanced by PLM to model the semantic relationship between TM sentences and the source sentence and obtain their context representation information simultaneously.

- apply the Copy Mechanism to alleviate the rare-word problem, especially when we do not have sufficient training corpus.

## 2 RELATED WORK

### TRANSLATION MEMORY

Our work aim at the studies that integrate Translation Memory into machine translation. Many methods have been proposed to combine TM and MT. For example, Koehn & Senellart (2010) applies the matched segments from TM to SMT in decoding. However, the integration of TM and NMT is more complicated, and limited efforts method is explored so far compared with the fusion of TM and SMT. Cao & Xiong (2018) identify this as a multi-input problem and use the multi-encode framework to encode the retrieved TM-target sentence and current source sentence. On this basis, Bapna & Firat (2019) propose a new approach that incorporates information from source sentence and TM-source while encoding the TM-target. Gu et al. (2017) encode all similar sentences from TM into context-vectors and use context-vectors to decode the target word by an additional attention mechanism. Xia et al. (2019) further extend the method of (Gu et al., 2017) in which they pack the sequential TM into a graph, leading to a more efficient attention computation. Our proposed method does not require an extra encoder, so no additional encoder parameters are introduced. Additionally, Different from the method proposed by Bapna & Firat (2019), we encode the three sentences (source, TM-source, TM-target) and obtain their context representation information simultaneously.

SOURCE DATA-AUGMENTED

Our work is related to studies that use similar sentences to improve translation quality. For example, Niehues et al. (2016) employ the pre-translations obtained by a phrase-based SMT system to augment the input sentences, and then concatenate the pre-translations and input sentences as the final input of the NMT model. This method is maintaining a separate SMT system which might introduce errors of its own. A similar principle, to dynamically adapt individual input sentences, Li et al. (2018) retrieve similar sentences from the training data to finetune the general NMT model. But this approach requires running expensive gradient descent steps before every translation. Xu et al. (2020) and Bulte & Tezcan (2019) employ the various experimenting to introduce how to retrieve similar sentences from TM and use the selected sentence as the extra information of source sentence. However, these studies only focus on how to enhance the context information of source sentences. How to use enhanced context information more effectively in the decoder is ignored. In our proposed methods, we try different ways to manipulate the extra context information from TM-source and TM-target.

EXTERNAL KNOWLEDGE FOR NMT

Our work is also related to previous works that incorporate external knowledge or information into NMT. Recently, a series of studies have explored the integration of SMT knowledge into the nmt system. For example, Zhou et al. (2017) and Wang et al. (2017) propose to integrate the SMT recommendations into NMT and improve the translation quality of NMT. Besides, there exist also studies on the learning of global context by the aid of discourse-level methods in document machine translation (Kuang & Xiong, 2018; Kuang et al., 2018; Zhang et al., 2018). Additionally, several successful attempts have been made to apply the pre-trained model for NMT (Ramachandran et al., 2017; Song et al., 2019; Weng et al., 2020). In addition to these, some studies also integrate external dictionaries into NMT or employ the force-decoding methods to constrain the decoding process of NMT by given words/phrases in target translations (Hokamp & Liu, 2017; Post & Vilar, 2018; Hasler et al., 2018).

## 3 PROBLEM FORMULATION

### 3.1 ATTENTIONAL NMT

Here, we will introduce neural machine translation based on the Transformer (Vaswani et al., 2017), which has achieved state-of-the-art performance in several language pairs. Formally, let $x = (\boldsymbol{x}_1, \boldsymbol{x}_2, ..., \boldsymbol{x}_I)$ be a source sentence and $y = (\boldsymbol{y}_1, \boldsymbol{y}_2, ..., \boldsymbol{y}_J)$ be a target sentence. The source and target sentence are parallel sentence pair. $\boldsymbol{x}_i$ denotes the $i$-th word vector representation in source sentence and $\boldsymbol{y}_j$ denotes the $j$-th word vector representation in target sentence.

By default, the encoder is composed of a stack of L identical layers. The output of $l$-th encoder layer can be computed as follows:

$$\boldsymbol{O}^l = \text{LN}(\text{ATT}(\boldsymbol{Q}^l, \boldsymbol{K}^l, \boldsymbol{V}^l) + \boldsymbol{H}^{l-1}) \tag{1}$$

$$\boldsymbol{H}^l = \text{LN}(\text{FFN}(\boldsymbol{O}^l) + \boldsymbol{O}^l)) \tag{2}$$

where ATT, LN and FFN are self-attention mechanism, layer normalization, and feed-forward networks with ReLU activation in between, respectively. $\boldsymbol{Q}^l, \boldsymbol{K}^l$ and $\boldsymbol{V}^l$ are the query, key and value matrix that are transformed from the $(l-1)$-th encoder layer output $\boldsymbol{H}^{l-1}$. We define the output of the $l$-th encoder layer as $H_c$

In the decoding stage, the decoder is also composed of a stack of L identical layers. the output $\boldsymbol{Z}^l$ of $l$-th decoder layer can be calculated as follows:

$$\boldsymbol{O}^l = \text{LN}(\text{ATT}(\boldsymbol{Q}^l, \boldsymbol{K}^l, \boldsymbol{V}^l) + \boldsymbol{Z}^{l-1}) \tag{3}$$

$$\boldsymbol{S}^l = \text{LN}(\text{ATT}(\boldsymbol{O}^l, \boldsymbol{K}_c, \boldsymbol{V}_c) + \boldsymbol{O}^l) \tag{4}$$

$$\boldsymbol{Z}^l = \text{LN}(\text{FFN}(\boldsymbol{S}^l) + \boldsymbol{S}^l)) \tag{5}$$

where $\boldsymbol{Q}^l$, $\boldsymbol{K}^l$ and $\boldsymbol{V}^l$ are transformed from the $(l-1)$-th decoder layer $\boldsymbol{Z}^{l-1}$. $\boldsymbol{K}_c$ and $\boldsymbol{V}_c$ are transformed from the $\boldsymbol{H}_c$. For self-attention mechanism $ATT$, the computed process is as follows:

$$\boldsymbol{c}_t = \sum_{j=1}^{J} \alpha_{tj} * \boldsymbol{v}_j \tag{6}$$

$$\alpha_{tj} = \frac{e_{tj}}{\sum_{k=1}^{J} e_{tk}} \tag{7}$$

$$e_{tj} = \boldsymbol{q}_t \boldsymbol{k}_j^T \tag{8}$$

where $\boldsymbol{c}_t$ is computed context vector of the $t$-th words, and the $\boldsymbol{q}_t$ is transformed from hidden representation of the $t$-th words, $\boldsymbol{k}$ and $\boldsymbol{v}$ is obtained from the previous layer of the decoder or the context representation of the encoder.

# 4 TRANSLATION MEMORY GUIDED NMT

In this section, we first introduce how TM sentence pairs are retrieved. Then describe the proposed Translation Memory Guided NMT (TMG-NMT) model in detail. Our TMG-NMT consists of two parts, a universal memory encoder for encoding context information of TM sentences and source sentence and a TM-guided decoder for using the TM context information to improve the translation quality.

## 4.1 SEGMENT-BASED TM RETRIEVAL

Given a source sentence $x$ to be translated, we find a matched example $(tx, ty)$ from a translation memory database $D = \{(x, y)\}_1^M$. Existing approaches proposed by previous work employ off-the-shelf search engines for the retrieval stage, and the retrieval source part $tx$ has the highest similarity score to the $x$. In general, when calculating similarity scores between $x$ and $tx$, the sentence level similarity score is employed, such as IDF-Based sentence score (Bapna & Firat, 2019), Fuzzy Match Score (Cao & Xiong, 2018), and Edit-Distance (Li et al., 2018).

Motivated by phrase-based SMT, in this paper, we propose the segment-based retrieval method that uses n-gram as the segment representation. When the retrieved sentence $tx$ has the same segment as the $x$, we can consider that $(tx, ty)$ may be helpful to improve the translation quality. Given a source sentence $x$, we denote its segments set as $s = (s_1, s_2, ..., s_k)$, where $s_k$ is one n-gram in the sentence $x$. For each $s_k$ in $s$, we try to find a matched example $(tx, ty)$ from $D$ what $tx$ contains the $s_k$. So, for sentence $x$, we can obtain a example set $\{(tx^i, ty^i)\}_1^N$ with $N$ sentence pairs. The advantages of the segment-based method are as follows:

- It doesn't need to compute the sentence level score, which greatly saves matching time.
- For $(x, y)$ pair in training data, we can find $N$ matched sentence pairs, this dramatically increases the size of our training corpus.

## 4.2 UNIVERSAL MEMORY ENCODER

Figure 1 illustrates the proposed universal memory encoder (UM-encoder). We concatenate the source sentence, TM-source sentence, and TM-target sentence as the input of the encoder. Formally, We use $tx = (\boldsymbol{tx}_1, \boldsymbol{tx}_2, ..., \boldsymbol{tx}_M)$ and $ty = (\boldsymbol{ty}_1, \boldsymbol{ty}_2, ..., \boldsymbol{ty}_N)$ to represent the retrieval TM-source sentence and TM-target sentence. The input of encoder $\boldsymbol{H}$ is concatenation of all vector representations of source words, TM-source words and TM-target words. $\boldsymbol{H} = [\boldsymbol{x}_1; ...; \boldsymbol{x}_i; ...; \boldsymbol{tx}_1; ...; \boldsymbol{tx}_m; \boldsymbol{ty}_1; ...; \boldsymbol{ty}_n]$.

Then, we split the $\boldsymbol{H}^l$ to get the source context $\boldsymbol{H}_x$, TM-source context $\boldsymbol{H}_{tx}$ and TM-target context $\boldsymbol{H}_{ty}$, as follows:

$$\boldsymbol{H}_x, \boldsymbol{H}_{tx}, \boldsymbol{H}_{ty} = \text{split}(\boldsymbol{H}^l, I, M, N) \tag{9}$$

where split is a function to split the matrix according to the length parameters $(I, M, N)$.

**Context control** Unlike the previous work, we don't employ additional context encoder or self-attention layer (Cao & Xiong, 2018; Kuang & Xiong, 2018) to encode and integrate contexts. And

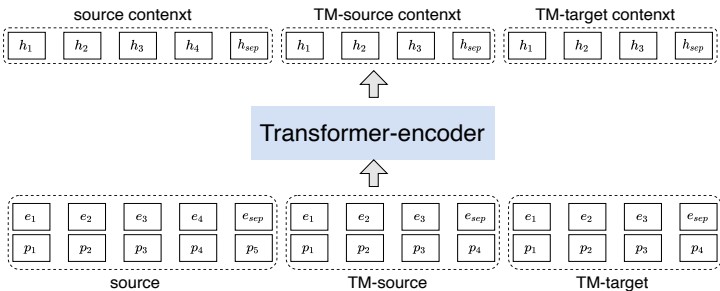

Figure 1: Architecture of the proposed UM-encoder model. A separation mark "SEP" is inserted between them. Position embedding is introduced to distinguish the word order in each sentence;

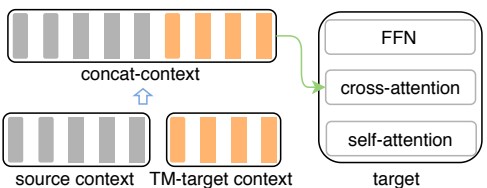

Figure 2: Architecture of TM-concat decoder

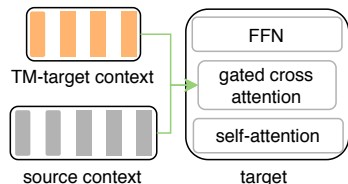

Figure 3: Architecture of TM-gate decoder

we only use one encoder to model the concatenated input. In this way, no additional parameters will be introduced and can also keep the same source sentence's semantic consistency. Meanwhile, to distinguish the context information from different sentences, a separation mark "SEP" is inserted between sentences. We arrange its position embedding separately to keep the word order for each sentence.

**Cross-lingual Encoding** The TM information contains two different languages: the source language and target language. So we need the UM-encoder to learn the ability to encode the semantics of different languages. For this goal, we introduce two methods into the UM-encoder:

- Firstly, we use a pre-trained cross-lingual language model (PLM) to initialize the UM-encoder's parameters, which provides an excellent cross-language modeling ability in advance.

- Secondly, We use TM-source as a bridge to establish the connection between the source sentence and TM-target. Because the TM-source and TM-target are the parallel sentence pair while the source sentence and TM-source are the same languages, it is easier to model the connection.

### 4.3 TM-GUIDED DECODER

By the UM-encoder, we can obtain the source context vectors, TM-source context vectors, and TM-target context vectors. So, we can employ the TM context information to guide the decoder to predict the next words. It is worth mentioning that we ignore TM-source context information because the words in the TM-source also appear in the source sentence, and we don't need to re-encode them while the words in TM-source but not in the source sentence are not helpful for translation. Additionally, we have used TM-source as a bridge to establish the semantic association between TM-target and source. There may exist different ways to use TM context information. We concentrate on the following four methods.

**TM-implicit Decoder** As we use UM-encoder for encoding, the source context already contains TM-target context information. So, we only use the source context $H_x$ as input $H_c$ of the decoder.

**TM-concat Decoder** Following Xu et al. (2020), we also concatenate the source context and TM-target context as the input of the decoder. Figure 2 illustrates the TM-concat decoder model. And

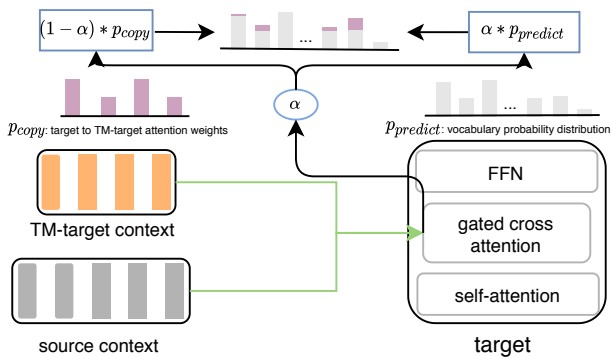

Figure 4: Architecture of TM-point decoder.

the $\boldsymbol{H}_c$ can be obtained as follows:

$$\boldsymbol{H}_c = [\boldsymbol{H}_x; \boldsymbol{H}_{ty}] \tag{10}$$

**TM-gate Decoder** Following Bapna & Firat (2019), We also try to use the gate attention mechanism for information fusion and use the $\boldsymbol{H}_x$ and $\boldsymbol{H}_{ty}$ as the decoder's input. Figure 4 schematically illustrates the TM-gate model. The $\boldsymbol{S}^l$ of the decoder are recalculated as follows:

$$\boldsymbol{S}_x^l = \mathrm{ATT}(\boldsymbol{O}^l, \boldsymbol{K}_x, \boldsymbol{V}_x) \tag{11}$$

$$\boldsymbol{S}_{ty}^l = \mathrm{ATT}(\boldsymbol{O}^l, \boldsymbol{K}_{ty}, \boldsymbol{V}_{ty}) \tag{12}$$

$$\boldsymbol{g}^l = \mathrm{sigmoid}(\mathrm{FFN}(\boldsymbol{S}_x^l, \boldsymbol{S}_{ty}^l, \boldsymbol{Z}^{l-1})) \tag{13}$$

$$\boldsymbol{S}^l = \mathrm{LN}(\boldsymbol{g}^l * \boldsymbol{S}_x^l + (1 - \boldsymbol{g}^l) * \boldsymbol{S}_{ty}^l + \boldsymbol{O}^l) \tag{14}$$

where $\boldsymbol{g}^l$ is the gated context vector obtained by sigmoid function and FFN. The $\boldsymbol{K}_x$ and $\boldsymbol{V}_x$ are transformed from the $\boldsymbol{H}_x$, and the $\boldsymbol{K}_{ty}$ and $\boldsymbol{V}_{ty}$ are transformed from the $\boldsymbol{H}_{ty}$.

**TM-pointer Decoder** We further integrate the pointer-network into the decoder to enhance the model's copy ability based on the TM-gate model. Figure 5 is a schematic representation of the TM-pointer model. At each decoding step $t$, we can get the current gated scalar $g_t$ by Equation (11) and attention distribution vectors $\boldsymbol{a}$ computed by Equation (10) from $l$-th decoder layer, where $\sum_0^N \boldsymbol{a}_i = 1$ and $N$ is the length of TM-target sentence. The final word prediction probability for $y_t$ is calculated as follows:

$$p(y_t|y_{<t}, x) = g_t * p_{predict}(y_t|y_{<t}, x) + (1 - g_t) * p_{copy}(y_t|y_{<t}, x) \tag{15}$$

where $p_{predict}(y_t|y_{<t}, x)$ is the vocabulary distribution computed by decoder. The $p_{copy}(y_t|y_{<t}, x)$ is the probability what copying a word from the TM-target sequence and we use the attention distribution vectors $\boldsymbol{a}$ represent this probability. Notice that if $y_t$ is not in the TM-target sentence, we set $p_{copy}(y_t|y_{<t}, x) = 0$.

## 5 EXPERIMENTS

We carried out a series of English-to-French translation experiments to evaluate the effectiveness of the proposed TMG-NMT and conducted in-depth analyses on experiment results and translations.

### 5.1 DATA

We used the following corpora in this work to validate the performance of the proposed model: the WMT training set (36M pairs), Open-Subtitles (33M pairs), the IWSLT bilingual corpus (237k pairs) and JRC-Acquis (797k pairs). For OpenSubtitles and JRC-Acquis, there is no publicly available test set. We create our own splits for validation and test. After segmentation, the JRC-Acquis test set contains 2000 sentence pairs and the OpenSubtitles test set has 2000 sentence pairs. For WMT, we

Table 1: BLEU scores on the English-to-French translation tasks and the baseline is trained using the concatenation of all the original corpus. OST indicate the OpenSubTitle test. Avg means the average scores on all test sets. Higher BLEU scores indicate better translation quality.

| | Model | IWSLT15 | OST | JRC | WMT14 | Avg | Δ |
|---|---|---|---|---|---|---|---|
| Baseline | Deep-Transformer | 43.07 | 34.76 | 54.31 | 42.02 | 43.57 | – |
| | Transformer-Bert | 44.25 | 35.18 | 56.43 | 43.53 | 44.84 | – |
| | w/ pretrain | 44.88 | 35.02 | 56.99 | 43.23 | 45.03 | – |
| TMG-NMT | TM-implicit | 44.85 | 35.57 | 61.31 | 43.94 | 46.41 | +1.38 |
| | TM-concat | 45.45 | 35.86 | 61.69 | 43.77 | 46.69 | +1.66 |
| | TM-gate | 46.18 | 35.44 | 61.71 | 43.97 | 46.82 | +1.79 |
| | TM-pointer | 46.97 | 36.49 | 62.60 | 44.64 | 47.67 | +2.64 |

use newstest 13 for validation and the newstest 14 for test. For IWSLT, we use test 2014 as validation and test 2015 as test.

In the training stage, for each sentence pair $(x, y)$ in the training corpus, we find up to $N$ similar sentence pairs $tx, ty$ from the training corpus. And we set the $N$ to 10. In the test stage, for each source sentence in the test set, we randomly select one of them when finding $N$ similar sentence pairs. Additionally, we restrict the retrieval set to the in-domain training corpus.

Following M-BERT (Pires et al., 2019), and for source vocabulary, we use a 110k shared WordPiece vocabulary. For the target vocabulary, we extract French-related character representations from the source vocabulary, and the target vocabulary size is 39000. We used the case-insensitive 4-gram NIST BLEU (Papineni et al., 2002) score as our evaluation metric.

## 5.2 EXPERIMENTAL SETTINGS

We implemented our TMG-NMT model based on a standard Transformer Base model (Vaswani et al., 2017). For the pre-trained cross-lingual language model, We take M-BERT trained on 104 languages as our first choice because its structure is consistent with the encoder of the Transformer.

For the encoder of the TMG-NMT, to conveniently load the parameters from M-BERT, we set the dimension of the hidden layer as 768 and set the size of the feed-forward layer is 3072. The number of layer and attention heads are 12 for the encoder. For the decoder of the TMG-NMT, the hidden dimension of all layers is set to 512, and the size of the feed-forward layer is set to 2048. We employ 8 parallel attention heads. The number of layers for the decoder is 6. Sentence pairs are batched together by approximate sequence length, and the maximum length of a sentence is limited to 100. We use label smoothing with value 0.1 and dropout with a rate of 0.1. We use Adam (Kingma & Ba, 2015) to update the parameters, and the learning rate was varied under a warm-up strategy with 6000 steps. Other settings of Transformer follow (Vaswani et al., 2017).

In order to compare the performance of our proposed model more fairly, we also use the same parameter settings to train the following two baseline systems. Our proposed models maintain the same configuration as Transformer-Bert.

- **Deep-Transformer** The baseline model that encoder contains 12 layers, hidden size is set to 512, the size of the feed-forward layer is 2048, and the number of attention head in the encoder are 12.

- **Transformer-Bert** The enhanced baseline model that the encoder has the same configuration with TMG-NMT. The encoder contains 12 layers, hidden size is set to 768, the size of the feed-forward layer is 3072, and the number of attention head in the encoder are 12.

## 5.3 EXPERIMENTAL RESULTS

**Main Results** Table 1 displays the translation performance measured in terms of BLEU scores. The results on the IWSLT15, OpenSubTitle, JRC and WMT14 indicate that every one of our proposed TMG-NMT models improves the translation accuracy in comparison to the transformer baseline. With respect to BLEU scores, we observe a consistent trend that the TM-gate model works bet-

Table 2: BLEU scores when the baseline model and TMG-NMT with TM-pointer model are only trained on WMT corpus.

| Model | IWSLT15 | OST | JRC | WMT14 | Avg | Δ |
|---|---|---|---|---|---|---|
| Transformer-Bert with pretrain | 39.32 | 22.33 | 52.59 | 42.76 | 39.25 | – |
| TM-pointer | 41.04 | 24.98 | 58.60 | 43.76 | 42.09 | +2.84 |

ter than the TM-concat model and TM-implicit model, while the TM-pointer model achieves the best accuracy over all test sets. On all test sets, the TM-pointer outperforms the baseline Deep-Transformer by 4.1 BLEU points and outperforms the other three proposed methods by 0.54∼1.26 BLEU points.

**compared with Enhanced baseline** We also trained an enhanced transformer model (Transformer-Bert), which has the same encoder parameters configuration as TMG-NMT. From Table 1, we can find that as the scale of parameters increases, Transformer-Bert achieves an average gain of 1.27 BLEU points over the Deep-transformer. When we use M-BERT's parameters to initialize the encoder, it further improves 0.19 BLEU points compared with Transformer-Bert. However, compared with the enhanced baseline, the proposed TMG-NMT still achieves an average improvement of 1.38∼2.64 BLEU points.

## 5.4 ADAPTATION

In the section, we consider a different translation scenario to NMT. In this scenario, we have a new TM that has never been seen when training the TMG-NMT model. Or the test sets we need to translate are a new domain, which is entirely different from the corpus for training the TMG-NMT model. So, we need to test whether the TMG-NMT model can be adapted to any new dataset by updating the retrieval TM database. In document-level translation tasks and online translation services, it's a common phenomenon that new sentence will be added continuously to the TM cache, and should be applied immediately to translate the following sentence.

So, following (Bapna & Firat, 2019), on the WMT training corpus, we train a baseline model and a TMG-NMT model. Then we valid the translation performance in JRC, IWSLT15, and OpenSubTitle test sets. We retrieval similar TM-source and TM-target from their respective training corpus when evaluating the adaptation of The proposed TMG-NMT.

Table 2 shows the corresponding results. Note that since the model is only trained on the WMT corpus, all test BLEU scores are lower than those shown in Table 1. The TM-pointer model significantly outperforms the base model on all test sets and achieves an average gain of 2.84 BLEU points. The results on three test sets indicate that the proposed methods show good adaptability to the unknown or new domain.

## 6 CONCLUSION

In this paper, we have presented TMG-NMT, an effective method to encode translation memory into NMT. Firstly, we explore a new framework that employs the universal memory encoder to simultaneously encode the TM information and source sentence, allowing the encoder to obtain the semantic relationship easily. Secondly, the TM-guided decoder is proposed to manipulate the information flow from TM to NMT decoder. Especially, we incorporate the pointer-network into the TM-guided decoder to further strengthen copying. Finally, We only use the n-gram matching algorithm to find similar sentences, making it easier for us to obtain the training corpus of the TMG-NMT model and expand the model's application scenarios.

Experiments on English to French translation shows that the proposed models can significantly improve translation quality. Further in-depth analysis demonstrate that our models show excellent adaptability to the unknown or new domain.

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
