# OpenReview forum: "Translation Memory Guided Neural Machine Translation"
_ICLR.cc/2021/Conference — Reject_

### Official Review · AnonReviewer4 · 2020-10-27
**The results are promising, but many parts of the work remained to be explored/reported.**

**Rating:** 4
**Confidence:** 5

**Review:**

** Summary **

(1) The authors proposed a translation system with an external memory. Given a sentence $x$ to be translated, they first retrieve a $(tx, ty)$ sentence pair from the training set through ``SEGMENT-BASED TM RETRIEVAL’’ defined in Section 4.1, where $tx$ and $ty$ are from source and target languages respectively. Then the $(x,tx,ty)$ are fused together to get the eventual translation, where authors design several ways to achieve that.

(2) Specifically, in the encoder side, the M-BERT model is leveraged to jointly encode the $x,tx,ty$.

(3) The improvement in Table 1 is significant.


** Clarity **
1.	Section 4.1 is unclear to me. In the 2nd paragraph of Section 4.1, ``For each s_k in s, we try to find a matched example (tx, ty) from D what tx contains the s_k’’: (1) There should be many sentence pairs (tx,ty) that tx contains s_k. Which ones should be kept? (2) What’is more, if tx contains s_k, can we say that the selected $tx$ is similar to $x$? (3) in experiments, the $n$ in  n-gram is set as? And what if we choose different $n$?
2.	Which script did you choose to evaluate BLEU score?

** Significance **
1.	The idea itself is not novel. Compared to the related work, the novel part of this work is: (i) a new retrieval way, which is not quite clear and convincing to me. (ii) a new way to aggregate multiple inputs (using M-BERT) and several different decoding methods. In experiments, there is no comparison with previous retrieval based methods. Similar idea also exists in [R3], which is missing from this paper. The differences with [R3] should be discussed.
2.	The authors did not provide what the retrieved sentences are like. Given a validation corpus (X,Y), and the corresponding retrieved (tX, tY), the authors should at least show the similarity between (X,tX), (Y,tY), which measures the retrieval quality.
3.	Please report the total training and total inference time, and make a comparison with standard Transformer model. Specifically, for Table 1, the inference time of each algorithm should be reported (retrieval time included).
4.	Why do you choose case-insensitive BLEU score for En->Fr, which is not commonly used in previous baselines.
5.	Considering the BERT is leveraged, you should discuss the relation with BERT + NMT [R1,R2].
6.     The authors should conduct experiments beyond English-to-French. More languages pairs should be verified.

Typos:
1.	compared with Enhanced baseline… -> Comparison with enhanced baseline


** Refereces **

R1: Zhu, Jinhua, Yingce Xia, Lijun Wu, Di He, Tao Qin, Wengang Zhou, Houqiang Li, and Tie-Yan Liu. "Incorporating bert into neural machine translation." ICLR’20, https://arxiv.org/pdf/2002.06823.pdf

R2: Yang, Jiacheng, Mingxuan Wang, Hao Zhou, Chengqi Zhao, Yong Yu, Weinan Zhang, and Lei Li. "Towards making the most of bert in neural machine translation." AAAI’20, https://arxiv.org/pdf/1908.05672.pdf

R3. Eriguchi, A., Rarrick, S. and Matsushita, H., 2019, November. Combining Translation Memory with Neural Machine Translation. In Proceedings of the 6th Workshop on Asian Translation (pp. 123-130).

---

### Official Review · AnonReviewer1 · 2020-10-28

**Rating:** 2
**Confidence:** 5

**Review:**

`The paper argues that the existing way of using Translation Memory (TM) in neural machine translation (NMT) is sub-optimal. Therefore it proposes TMG-NMT, namely Translation Memory Guided NMT, which consists of two parts, a universal memory encoder and a TM guided decoder. Experiments are performed to demonstrate that their method can significantly improve the translation quality and show strong adaptation for a new domain.

Pros: None

Cons:

1. The main concern of this paper is that, the contributions are quite limited. The authors claimed three contributions: n-gram retrieval, universal encoder, and using the copy mechanism. Basically, none of them is novel.
  - Bapna & Firat (2019) and Xu et al. (2020) have used n-gram matching for retrieval.
  - In late 2020, what is the novelty of using a multilingual BERT when encoding source sentences and retrieved TM sentences? Very little if any.
  - Likewise, using the copy mechanism to tackle rare word problems seems a regular approach.

 Overall, I don't see any of these so-called contributions are truly technically original. This paper seems a very hurry combination of some existing techniques. I basically learn nothing new from reading this submission.

2. Another one of the key concerns about the paper is the lack of rigorous experimentation to study the usefulness of the proposed method. Despite the paper stating that there have been earlier work (Gu et al. 2017, Can & Xiong. 2018, Xia et al. 2019) that explore Translation Memory in NMT, the paper does not compare with them and only compare to non-TM-guided baselines, making the improvement less convincing. In addition, what is the language pair evaluated in this paper, which was not even mentioned...

3. Considering the limited results, a deeper analysis of the proposed method would have been nice. Is the semantic relationship between the source sentence and TM sentences well learned in TMG-NMT? What kind of translation error can be well addressed with the help of TM? Further analysis of the proposed model would provide greater insight to the community.

4. Section 5.4: The results would have been more complete if another setting is considered where the transformer is adapted to the target domain without using the TM mechanism, such as fine-tuning the vanilla transformer on the provided TM parallel sentences. In this way, the adaptation ability of TMG-NMT could be better proven.

5. To be honest, the writing of the current version seems a disaster. Not to mention the impressive amount of grammar errors, many parts of the storytelling are logically incoherent. For example,
  - "Although Bapna & Firat (2019) and Xu et al. (2020) **also** use the n-gram method to search, they still need to select the corresponding sentence that maximizes the n-gram similarity with the source sentence. " - The usage of "also" is so weird, when you didn't even mention you are using n-gram method in advance... And by the way, what is the difference between your ngram matching and theirs? You should've made it clear.
  - "To obtain sufficient training corpus and train network parameters more fully and effectively, in this paper, we also modify the retrieval algorithm and use a pre-trained language model (PLM) to initialize the encoder’s parameters. Partially inspired by phrase-based SMT, we don’t compute the sentence level similarity score between two sentences in our retrieved method. If two sentences have a common n-gram segment, we assume that they are similar, and the sentence pairs of the TM database can provide a useful segment to help improve the translation quality. Currently, many studies have proven that PLM can offer valuable prior knowledge to enhance the translation performance of NMT (Weng et al., 2020; Song et al., 2019), So we also employ PLM to initialize the parameters of the encoder and give encoder well-trained parameters as a starting point." - What is the logical relationship b/w the first sentence and the second one?
  - Please check minors for more details.


*****
Minors:
1) It would have been nice to see that the format of the reference are unified.
2) Khandelwal et al, 2020 [1] propose a novel way to incorporate Translation Memory into NMT which may bring you more thoughts towards using TM.

Typos: too many.
1. Equation (2): a redundant close paren
2. Section 3.1 penultimate paragraph: l-th encoder layer -> L-th encoder layer
3. Section 4.2 First paragraph: N->M n->m

Grammar errors:

Too many. E.g., In the contribution part in the intro, the second and the third items start with "does" and "apply". What are the subj of these two verbs?

Please try to properly use Grammarly to check your writing before submission.


[1] "Nearest Neighbor Machine Translation. Khandelwal et al. 2020 arXiv."


******
Reasons for score:
The novelty of this paper is basically none. The experimental results are limited and the comparison with prior work is none, which cannot fully demonstrate the effectiveness of the proposed method.

---

### Official Review · AnonReviewer3 · 2020-10-29
**Relevant and interesting focus on TM use in NMT; a number of issues to clarify**

**Rating:** 4
**Confidence:** 4

**Review:**

This paper describes several improvements on using information from a Translation Memory (TM) in Neural Machine Translation (NMT). In the spirit of several prior work, the approach relies on 1) a retrieval step to obtain TM content that is related to the current source sentence to translate, 2) an encoder combining the source sentence with retrieved TM content, and 3) a decoder using the joint encoded information to produce a (target) translation. Experiments are conducted on benchmark French-English data, showing consistent improvement over classical baselines.

Translation Memories are important Computer-Aided Translation (CAT) tools, likely the most widely used CAT tools by translation professionals and agencies. As such, it is important to study how they can be used to improve translation quality for example through inclusion in NMT. This study is therefore a welcome addition to the relatively limited work investigating this topic. I personally wish there was more work on integrating existing translation resources in MT. However, the experimental setup does not really correspond to a typical TM use. It essentially leverages the idea of reusing close matches to a source sentence in the training data. The idea is interesting, but only loosely related to TM. On the other hand, the experiments in Section 5.4 are much closer to a TM, it is unfortunate that these experiments are quite limited.

Despite the general relevant and interesting focus of this work, there are a number of issues discussed below, related mainly to modeling and to the experimental evaluation.

MODELLING:
Each of the three components on the method (retrieval, encoding, and decoding) introduces some novelty. There are also a number of issues to resolve.

1) The retrieval (sec. 4.1) uses an n-gram matching technique, which is contrasted with the usual computation of sentence similarities (edit distance, fuzzy match or idf-based score). I basically don’t buy the advantages put forward in the paper:
- The cost of retrieval in a TM is dominated by the requirement to go over the entire memory for each source sentence, not by the computation of the score. The n-gram matching would still incur that cost, unless some smart way to retrieve matching sentences (such as an inverted n-gram index) is implemented. Unfortunately there is no detail at all in the paper about how the ngram matching is performed.
- The fact that one can retrieve N matched sentence pairs for each (x,y) pair in the training data is no different from retrieving the top-N sentence pairs using any of the usual similarity metrics.

Additionally, when retrieving N>1 sentence pairs from the TM, it is not entirely clear how the N pairs are used. One interpretation of the second paragraph of 5.1 is that this actually yields N different training instances at training time, while one match is randomly picked at prediction time. This should be clarified. In addition, this would introduce additional randomness at prediction time, producing possibly different target translations. It would be good to assess the impact of this choice on the performance, and compare against the obvious choice of picking the closest match.

2) The encoding is straightforward but clever. It is not entirely clear how the encoder keeps track of the split of the context into I+M+N (one assumes here that N is no longer the number of matched pairs, but the length of the TM-target context) — is it through propagating the separation marker, hardcoded in the encoder, through some other way?

3) The decoding is done in four different manners, offering various ways to integrate the TM-target information into the prediction. However, the description of « TM-pointer Decoder » in Section 4.3 seems faulty: Eq. 11 shows how to get S_x^l through the self-attention mechanism and Eq. 10 illustrates the concatenation mechanism in the TM-concat decoder, they can’t help get g_t and the attention distribution vector a.

EVALUATION:

Strengths:
+ Shows consistent non trivial gains
+ Uses a large corpus, with several domains
+ Interesting « domain adaptation » mode shows good results [Sec. 5.4]

Weaknesses:
- Limited to French-English (close languages with lots of cognates, lots of resources, high performance) — it would be interesting to show how this works on radically different languages, especially in a lower resource setting where an existing TM may greatly help.
- Limited comparison to a couple baselines. None of the methods cited in the related work is tested against.
- It is unclear what significance test was used, if any, to back the claim of « significantly outperforms » (e.g. end Section 5.4).

Small typos and clarifications:
« by reuse existing » (Sec 1)
« nmt » (Sec 2) -> NMT
«  Formally, We » (Sec 4.2)
What is the sentence after Eq. 15 (« The p_copy… ») trying to tell us. This is not clear.
« we set the N to 10 » (Sec 5.1)
« set the size … is » (Sec 5.2)
« we valid the translation performance » (Sec 5.4)

---

### Official Review · AnonReviewer2 · 2020-10-29
**Translation Memory and retrieval from it is not described**

**Rating:** 4
**Confidence:** 4

**Review:**

This paper presents a way to integrate a translation memory into a neural machine translation model. They introduce a novel method for finding matching sentences (which they do not properly describe) and they also have 4 methods from including this information into an NMT model, with the best approach using multiple sources combined with gated attention and a pointer network for copying rare words. They show improvements of up to 2.64 BLEU over a reasonable but not SOTA baseline for English-French task.

My major problem with this paper is that the model and the experiments are not adequately described.

The segment based retrieval function is one of the two core contributions of the paper, but it cannot be fully described. They claim that any sentence in the TM that matches any n-gram in the sentence x. This would result in a huge set of matching sentences, where 1-gram that matches could be the work "the". So there must be a way of ordering the matches or scoring the matches to result in a good set of N matches where N=10 here. How do they do this?

At no point to they describe where they get the translation memory from and/or how big it is. This is a fundamental part of the model. I have searched and can't find this - did I miss it?
I find this to be an extremely worrying omission - especially as there should be a discussion about the properties of this TM and how it compares to for instance using a phrase-table/lexicon etc.

Their translation baselines are reasonable but probably significantly lower than SOTA (44.6 BLEU vs highest published 45.6 on WMT14 from http://nlpprogress.com/english/machine_translation.html)
They are not directly comparable however because they should be reporting detokenized BLEU using sacreBLEU. Their scores are tokenised lowercase text which means that their numbers are artificially elevates when compared to true cased tokenised text.

---

### Decision · Program_Chairs · 2021-01-07
**Final Decision**

**Decision:**

Reject

**Comment:**

This paper presents a way to use a translation memory (TM) to improve neural machine translation.  Basically the proposed model uses a n-gram retrieve matching sentence （or pieces） and takes advantage of the useful parts using gated attention and copying mechanism.  Although the idea of leveraging TM in the context of NMT is not new,  this work seems to be a fair contribution. My major concerns are the following
1. The retrieval part  is not clearly presented, raising questions about  complexities and the noise brought by the common words. The authors should give a better exposition on the ranking mechanism.
2. The experiments are not convincing enough since the proposed model is not compared to the SOTA and the competitive models described in the prior work.

In conclusion I would suggest to reject this paper.